# Identification of *Contracaecum rudolphii* (Nematoda: Anisakidae) in Great Cormorants *Phalacrocorax carbo sinensis* (Blumenbach, 1978) from Southern Italy

**DOI:** 10.3390/vetsci10030194

**Published:** 2023-03-04

**Authors:** Gaetano Cammilleri, Stefano D’Amelio, Vincenzo Ferrantelli, Antonella Costa, Maria Drussilla Buscemi, Annamaria Castello, Emanuela Bacchi, Elisa Goffredo, Maria Emanuela Mancini, Serena Cavallero

**Affiliations:** 1Centro di Referenza Nazionale per le Anisakiasi, Istituto Zooprofilattico Sperimentale della Sicilia “A. Mirri”, Via Gino Marinuzzi 3, 90129 Palermo, Italy; 2Department of Public Health and Infectious Diseases, Sapienza University of Rome, Piazzale Aldo Moro 5, 00185 Rome, Italy; 3Istituto Zooprofilattico Sperimentale della Puglia e della Basilicata, Via Manfredonia 20, 71121 Foggia, Italy

**Keywords:** *Anisakidae*, *Contracaecum*, PCR-RFLP, South Mediterranean

## Abstract

**Simple Summary:**

Four dead *Phalacrocorax carbo sinensis* specimens from Southern Italy coasts were examined for *Contracaecum* sp. detection. 181 *Contracaecum* sp. larvae and adults were found in the viscera of all the *P. carbo* sinensis examined. The PCR-RFLP analysis showed the presence of *Contracaecum rudolphii* A and B. A co-infestation of *C. rudolphii* A and B was found in *P. carbo sinensis* from Leporano Bay. This study provides a first report of the presence of *Contracaecum* sp. in *P. carbo sinensis* from Southern Italy.

**Abstract:**

In this study, four dead great cormorant *Phalacrocorax carbo sinensis* (Blumenbach, 1978) specimens, collected from the coasts and lakes of Southern Italy, were examined by necropsy for the detection of *Contraceacum* sp. The adults and larvae found were subjected to morphological analysis and molecular identification by PCR-RFLP. A total of 181 *Contracaecum* specimens were detected in all of the four great cormorants examined (prevalence = 100%), showing an intensity of infestation between nine and ninety-two. A co-infestation by adult and larval forms of *Contracaecum rudolphii* was found only in one of the great cormorants examined. Following molecular investigations, 48 specimens of *C. rudolphii* A and 38 specimens of *C. rudolphii* B were detected, revealing co-infestation solely for the great cormorant from Leporano Bay (Southern Italy). Our results showed an opposite ratio between *C. rudolphii* A and *C. rudolphii* B in Pantelleria and in Salso Lake (Southern Italy) compared to what was reported in the literature, probably due to migratory stopovers and the ecology of the infested fish species, confirming the role of *Contracaecum* nematodes as ecological tags of their hosts.

## 1. Introduction

The Anisakidae family includes fish-borne nematodes with zoonotic significance and worldwide distribution [1,2,3]. Among the Anisakidae family, *Contracaecum* spp. nematodes occur as adults in the stomachs of different piscivorous birds, including pelicans and cormorants [4]. The genus includes organisms with a complex life cycle, having aquatic invertebrates as first intermediate hosts (small crustaceans, polychaetes, and gastropods) [5], whereas fish represent the second intermediate or paratenic hosts. In particular, *Contracaecum rudolphii* (s.l.) (Hartwich, 1964) is a complex of five sibling species of anisakid nematodes discovered across the world through the application of molecular analysis showing different genetic structure, geographical distribution, host preference, and life cycle [3]. In the Mediterranean, the presence of such species as *C. rudolphii* (Hartwich, 1964) (s.l.) was documented in fish residing in both brackish and freshwater habitats, with a higher occurrence of *C. rudolphii* A found in brackish water habitats [6,7] and *C. rudolphii* B in freshwater environments [4].

The great cormorant, *Phalacrocorax carbo* (Blumenbach, 1978), belongs to the Phalacrocoracidae family. *P. carbo sinensis* is a highly specialized ichthyophagous bird, which it captures during the day by diving. It is a polytypic species with a sub-cosmopolitan distribution and a breeding population reaching 20,900 pairs in Central Europe and central-eastern Mediterranean areas [8]. The *P. carbo sinensis* subspecies has been found near coasts, estuaries, lagoons, as well as large rivers and lakes in inland areas of Sweden, Belarus, Poland, Germany, the Netherlands, France, Spain, Croatia, Northern Greece, and Northern Italy [8]. Between October and April, *P. carbo sinensis* migrates and winters in Italy, where very few individuals stay also during the summer months [9]. Most of the nesting colonies are located in the freshwater wetlands of Northern Italy and in the lagoons along the Adriatic coasts [8]. Small nesting colonies were present in Sardinia and Sicily from 1960 until the beginning of 1990 [10].

Different studies on the parasite fauna of the great cormorant in the Mediterranean confirmed that this species is the main host of *C. rudolphii* s.l. nematodes, showing very high intensity of infestation values and a prevalence of 100% [11]. However, most of the studies available in the literature are focused on the Northern Mediterranean, including the coasts of Sardinia [4,6,12,13]. As far as we know, the occurrence and molecular identification of *C. rudolphii* s.l. adults in South Mediterranean waters have been poorly investigated, requiring further examination.

At present, the molecular identification of *Contracaecum* s.l. at species level is carried out by PCR-RFLP analysis and sequencing of the internal transcribed spacer (ITS) nuclear locus and mitochondrial loci, such as the cytochrome c oxidase 1 (*cox1*), cytochrome c oxidase 2 (*cox2*), and the small subunit of rRNA (*rrnS*) [14].

This study aimed to deepen knowledge of the occurrence and molecular identification of *Contracaecum* spp. in cormorants collected along the coasts and in the inland of Southern Italy, in order to enrich the scarce amount of data available about this host in the aforementioned geographical area.

## 2. Materials and Methods

### 2.1. Samples Collection

A total of four specimens of *P. carbo sinensis,* the Eurasian subspecies of the great cormorant, were collected for the detection of *Contracaecum* spp. All the specimens were found dead along the coastlines and the waterways of Southern Italy and were supplied by the local authority (Figure 1).

In particular, one individual was obtained from the staff of the Wildlife Rescue Center of Cattolica Eraclea (Sicily, Southern Italy) near the Platani River in 2011. The Platani is one of the most important watercourses of the southern side of Sicily, with a 103 km length and a basin width of 1785 km^2^. The river has a torrential character with considerable floods in autumn and strongly low levels in summer. A second cormorant was collected from Pantelleria coasts in 2013, following a report from the coast guard. Pantelleria is the largest volcanic satellite island of Sicily, with an area of 83 km^2^. It is located 120 km southwest of Sicily, 70 m east of the Tunisian coast, and has no waterways. The other two cormorants were collected along the coasts (Leporano Bay, Taranto) and waterways (Salso Lake, Gargano Park, Manfredonia) of the Apulia region in 2017 and 2022, respectively. Leporano Bay is located in the Ionian Sea; it has a coastline consisting of rocky shore and several bays with sandy beaches. Salso Lake is a stretch of water of about 550 hectares that receives its waters mainly from the Roncone canal connected directly to the Cervaro stream, while the Candelaro stream divides this area from the west side along with the Frattarolo swamp, a wet area that is decidedly more brackish.

### 2.2. Visual Inspection and Morphological Analysis

All the cormorants were subjected to necroscopy and visual inspection of the abdominal cavity for the detection of nematode parasites. All the nematodes collected were repeatedly washed in physiological saline solution and stored with 70% ethanol until morphological analysis. The anterior and posterior ends of the nematodes were cleared with glycerine and lactic acid–phenol (1:1) for morphological identification by light microscopy Leica DM 2000 (Wetzlar, Germany), following the taxonomic keys based on the morphology of lips and interlabial tips, length of spicule, and morphology of the spicule tip [15,16]. The remaining parts of a selection of individual worms were subjected to molecular characterization.

### 2.3. Molecular Analysis, Data Collection, and Analysis

Genomic DNA was isolated using the ISOLATE II Genomic DNA Kit (Meridian Bioscience) following the manufacturer’s protocol and then used for PCRs. The diagnostic key based on PCR-RFLP analysis of *rrnS* and ITS markers was performed according to D’Amelio et al. (2007) [14] and PCRs conditions and primers were used according to D’Amelio et al. (2007) [14]. In particular, the mitochondrial *rrnS* positive amplicons were digested with *Rsa*I and *Dde*I endonucleases and the positive nuclear ITS amplicons were digested with *Tsp*509I endonuclease. Fragments obtained were resolved by electrophoresis in 2% agarose gels, stained with SybrSafe (Invitrogen, Waltham, MA, USA), detected upon transillumination, and the sizes of fragments were characterized by comparison to a 100 bp DNA ladder as size marker (Promega, Amsterdam, the Netherlands). The prevalence, mean intensity, and abundance parameters were inferred using Quantitative Parasitology 3.0 software [17].

## 3. Results

All specimens of *P. carbo sinensis* were found to be infected by *C. rudolphii* (s.l.) (prevalence = 100%) at larval and/or adult stages. A total of 181 nematode specimens were detected by necroscopy analysis. In particular, 92 specimens were found in the stomach content of the cormorant collected along the Platani River; the two cormorants collected at Salso Lake and at Leporano Bay (Apulia, Southern Italy) showed the presence of nine specimens in the stomach and 50 specimens in the intestine, respectively. Furthermore, 30 specimens were detected in the stomach of the cormorant from Pantelleria. The infestation parameters of the samples examined are reported in Table 1.

The cormorant from the Platani River showed the highest intensity of infestation (92).

A co-infestation by adult and larval forms of *C. rudolphii* was found in the cormorant from Platani River. The intensity of infestation of adult forms ranged between nine and fifty, showing an M:F sex ratio of 0.6:1. Only the cormorant from Leporano Bay showed a greater number of males than females.

The morphological analysis confirmed the presence of 54 fourth-stage larvae and 127 adults of *C. rudolphii* sensu (Hartwich, 1964). Adults showed a brownish-yellowish and transversely striated cuticle, lips divided into two lobes, and interlabia well-developed and bifurcated in the distal end (Figure 2). Males showed two subequal spicules with a length in the range of 4–10 mm, according to the morphological key [15].

All the nematodes selected as a subsample for the four cormorants included in this study returned PCR amplicons of the size expected from the target loci (~1000 bp and ~600 bp for ITS and *rrnS*, respectively). According to the RFLP patterns obtained for the ITS and *rrnS* genomic regions, the 35 nematodes belonging to the subsample for the cormorants from Pantelleria and Platani River (Sicily, Southern Italy) were identified as *C. rudolphii* B; the subsample for the cormorant from Leporano (Taranto, Southern Italy) included both *C. rudolphii* A (*n* = 31) and B (*n* = 3) as concurring parasites; and the 17 nematodes belonging to the subsample for the cormorant collected from Salso Lake were identified as *C. rudolphii* A. Representative gel images displaying the RFLP profiles of *C. rudolphii* A and B are reported in Figure 3, Figure 4 and Figure 5.

## 4. Discussion

Several studies describe *C. rudolphii* as a common anisakid parasite of fish-eating birds with worldwide distribution. Regarding the cormorants, to date *C. rudolphii* (s.l.) has been detected outside Europe, in particular in *P. brasilianus* in Chile [18] and in *P. auritus* in the USA [14]. Regarding European regions, it was reported in *P. carbo* in the Czech Republic, Poland, and Germany [19,20,21] and in *P. aristotelis* in Spain and also in Italy [4,7]. This study provided information regarding the members of *C. rudolphii* complex parasites of cormorants in a poorly investigated area, such as the south of Italy. The prevalence of *Contracaecum* observed in this study is comparable to evidence for *P. carbo sinensis* from Central and Northern Italy [4,12]. The results of this study on the occurrence of the two common taxa (A and B) of *C. rudolphii* s.l. seem to confirm the distribution pattern observed also in other studies focused on Mediterranean cormorants, except for the cormorants collected from Pantelleria and Salso Lake. Indeed, previous studies reported a greater presence of *C. rudolphii* A in brackish-marine waters, whereas *C. rudolphii* B was found mostly in freshwater habitats [4,22]. Similar evidence was collected from sedentary and wintering cormorants (*P. carbo sinensis*) from the pre-mountain area of the Alps in Northern Italy, an important crossroads for most of the bird migration routes.

The sibling species *C. rudolphii* B was previously shown to be the most widespread taxa, constituting the largest sibling fraction as the endemic species, while cormorants originating from the breeding brackish lagoons and marine coastal environments of Northern Europe were infected by *C. rudolphii* A, probably acquired from their breeding sites or migration stopovers [12]. In this context, the physical properties of the aquatic ecosystems seem to contribute to maintaining the different occurrence of the *C. rudolphii* s.l. species. The cormorants collected from Pantelleria and Salso Lake revealed the exclusive presence of *C. rudolphii* B and *C. rudolphii* A, respectively. Our findings on the cormorant collected from Salso Lake are in agreement with evidence from Carmeno et al. (2022) [12] on *Contracaecum* collected at Como Lake, showing a higher prevalence of *C. rudolphii* A detected, even as a single infection. This reverse trend can be traced back to the presence of wintering individuals experiencing different feeding behaviors during their seasonal migration cycles [12,23]. In Southern Italy, the wintering population of cormorants reaches far greater numbers than the breeding population [9,24]. Furthermore, recent censuses did not report the presence of nesting colonies in Sicily since the early 1990s [10]. The absence of *C. rudolphii* A in the cormorant from Pantelleria suggests that *C. rudolphii* B infected the host in environments other than those of the studied area [6]. Considering that *P. carbo sinensis* is present exclusively as a wintering species in Sicily, it could be possible that *C. rudolphii* B infected its host in the breeding areas of Central and Northern Europe, where L3 larvae of this species were detected in fish [4,25]. Small nesting colonies of this cormorant species were present in Sicily only from 1960 until the beginning of 1990 [10]. Further studies based on the analysis of oral pellets [26] of cormorants wintering in Sicily could be useful to confirm this hypothesis. Conversely to the cormorant from Pantelleria, the opposite ratio of infestation taxa found in the cormorant from Salso Lake could be related to the hydrogeological aspects of the site of collection, consisting of a brackish marsh characterised by the presence of *Anguilla anguilla,* recognised as one of the species most commonly infected by *C. rudolphii* A [4,27].

Even the sampling period could be an additional clue supporting this hypothesis. It was proven that from December to March, cormorants can be infected by L3-stage larvae of *C. rudolphii* (s.l.) in situ by feeding out of fish in wintering sites of Central Italy [4]. Other data available from Italy regard Sardinia and central and northern regions. So far, Amor et al. (2020) have reported the coexistence of *C. rudolphii* A and *C. rudolphii* B in Sardinia and a rare heterozygote of these species. Furthermore, in Central Italy, cormorants from brackish and freshwater ecosystems appeared to be infected by both taxonomic units A and B of the *C. rudolphii* complex [4]. Co-infestations between *C. rudolphii* A and *C. rudolphii* B were also reported by Szostakowska and Fagerholm (2012) [11] in brackish water habitats of Northern Europe.

In this study, co-infestation with two species, A and B, was found only in the cormorant from Leporano Bay, showing a higher occurrence of *C. rudolphii* A (91%) compared to *C. rudolphii* B (9%) in accordance with what was found by Mattiucci et al. (2020) [4] in brackish habitats from Central Italy. Our findings seem to confirm the role of the definitive hosts and their wintering behavior in the heterogeneity of the infection with the two *Contracaecum* species [4,14]

## 5. Conclusions

This study provides new data on the occurrence and prevalence of *Contracaecum* spp. in *P. carbo sinensis* collected from aquatic environments of Southern Italy. As far as we know, this work reports the first data on the presence of *C. rudolphii* species complex in great cormorants from South Mediterranean coasts. Our results confirmed the usefulness of *rrnS* and ITS markers for species identification of the Anisakidae parasites [14,28,29] and the potential role of *C. rudolphii* s.l. as ecological tags to deepen knowledge of the ecology and migratory behavior of their definitive hosts.

## Figures and Tables

**Figure 1 vetsci-10-00194-f001:**
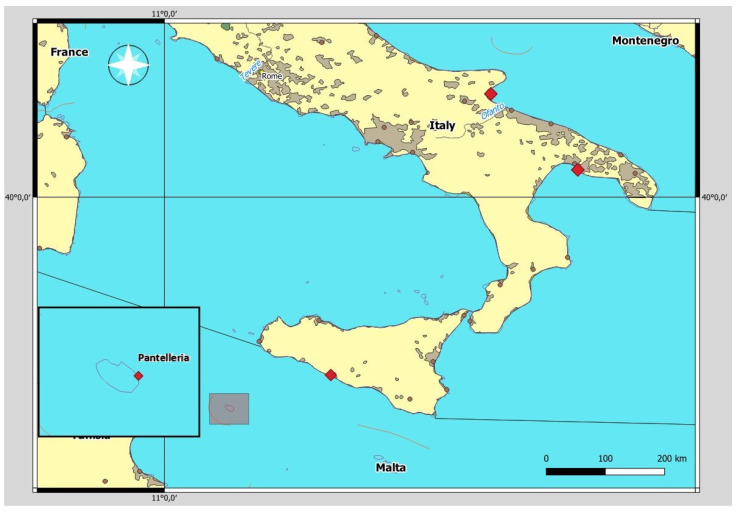
Map of the sites where the great cormorants (*P. carbo sinensis*) examined were collected.

**Figure 2 vetsci-10-00194-f002:**
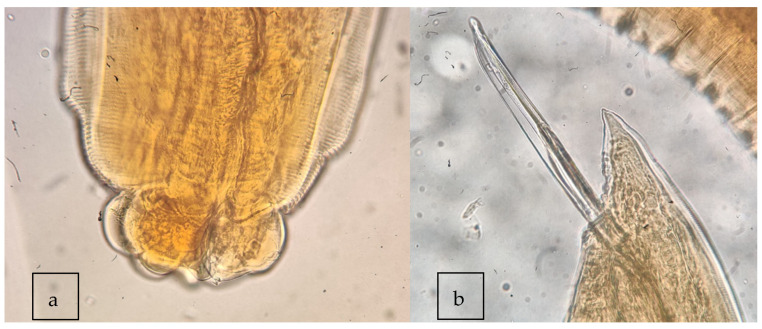
Photomicrographs of the *Contracaecum* adults found. (**a**) Mouth region of the adult form; (**b**) Tail region of the male adult form.

**Figure 3 vetsci-10-00194-f003:**
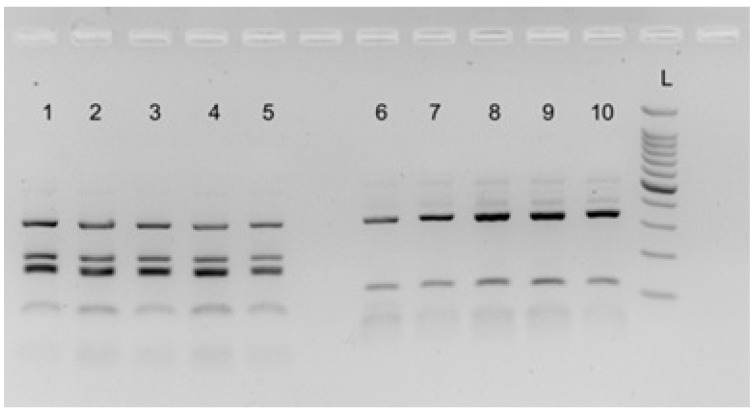
A representative gel displaying the Restriction Fragment Length Polymorphism profiles of *C. rudolphii* A obtained after digestion of ITS amplicons with endonuclease *Tsp509*I (lanes 1–5) and of *rrnS* amplicons with *Rsa*I (lines 6–10). L: molecular ladder 100 bp (500 bp lane is highlighted).

**Figure 4 vetsci-10-00194-f004:**
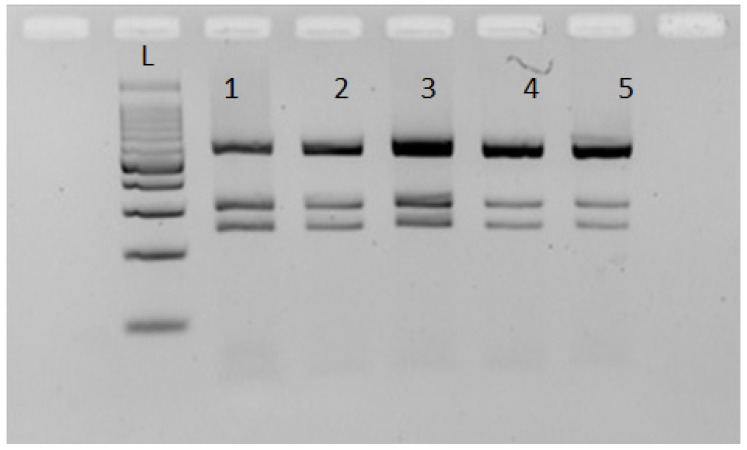
A representative gel displaying the Restriction Fragment Length Polymorphism profiles of *C. rudolphii* A obtained after digestion of *rrnS* amplicons with *Dde*I. L: molecular ladder 100 bp.

**Figure 5 vetsci-10-00194-f005:**
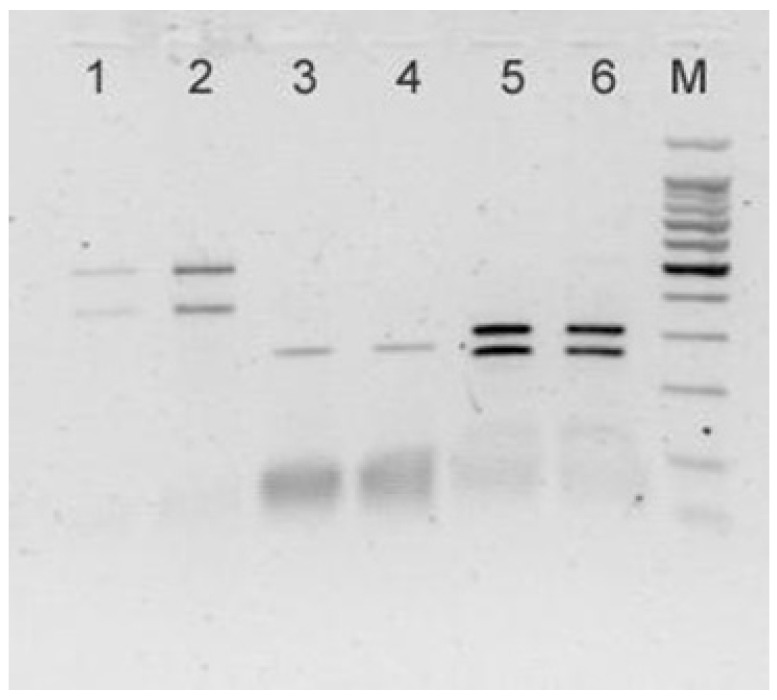
A representative gel displaying the Restriction Fragment Length Polymorphism profiles of *C. rudolphii* B obtained after digestion of ITS amplicons with endonuclease *Tsp509*I (lines 1, 2) and of *rrn*S amplicons with *Rsa*I (lines 3, 4) and *Dde*I (lines 5, 6). M: molecular ladder 100 bp.

**Table 1 vetsci-10-00194-t001:** Infestation parameters of the *C. rudolphii* (s.l.) specimens isolated from *P. carbo sinensis* collected in the sampling areas considered (N: number of specimens).

Sampling Area	N Birds	N. *Contracaecum* *rudolphii* (s.l.)	Location (N)	N Larvae	N Adults
Cattolica Eraclea (Platani River)	1	92	Stomach (92)	54	38
Pantelleria	1	30	Stomach (30)	-	30
Leporano Bay	1	50	Stomach (35), Intestine (15)	-	50
Salso Lake	1	9	Stomach (9)	-	9

## Data Availability

Data are not available for the type of study conducted.

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
