# Peer review of "Identification of Contracaecum rudolphii (Nematoda: Anisakidae) in Great Cormorants Phalacrocorax carbo sinensis (Blumenbach, 1978) from Southern Italy"

_vetsci, 2023, doi:10.3390/vetsci10030194_

Round 1

Reviewer 1 Report

The manuscript concerns on the occurrence of the nematode Contracaecum rudolphii in great cormorant in Italy. The data obtained are interesting, however, the paper  needs some major revisions.

 1. Title: since nematodes are identified to a species, then you should use this name (Contracaecum rudolphii) in the title. Specifying "Contracaecum sp." is pointless.

 2. Simple summary, Abstract: see comment 1.

The abbreviation "sp." stands for singular (one unidentified species); the abbreviation "spp." means plural, i.e. at least two species. Please rephrase these two chapters accordingly.

 3. Lines 39-40: „Among the Anisakidae family, Contracaecum spp. nematodes occur as adults in the stomach of different piscivorous birds, including pelicans and cormorants [4]. The complex life cycle of Contracaecum rudolphii s.l. includes aquatic invertebrates as first intermediate hosts (small crustaceans, polychaetes and gastropods)” – not clear – These two sentences have no connection with each other - first the genus "Contracaecum" is mentioned and then suddenly the species "C. rudolphii" appears. It would have been appropriate to write that this species of nematode occurs in the cormorant.

 4. Lines 41-42: „… invertebrates as first intermediate hosts (small crustaceans, polychaetes and gastropods) [5],…” – incorrect citation, the authors of this publication Kanarek and Bohdanowicz (2009) did not study the life cycle of C. rudolphii.

 5.Lines 43-45: „Mediterranean, the presence of C. rudolphii s.l. was documented in fish …[6,7]” - incorrect citation, authors of publication [7] studied fish only from Poland.

 6. Lines 74-76: „A total of four specimens of Phalacrocorax carbo sinensis, the Eurasian subspecies of the Great Cormorant, were collected from 2011 to 2022” – strange notation - four cormorants were collected in 12 years ? Please remove this sentence, and add the year data to the next paragraph (to specific cormorants).

 7. Lines 101-103: „ …following the taxonomic keys based on the morphology of lips and interlabial tips, length of spicule and morphology of the spicule tip [15]: - incorrect citation, in this article [Li et al., 2005] there is no description of this nematode species. .

 8. Line 120: since the nematode species was identified, it must be stated that C. rudolphii was found; the name "Contracaecum spp." suggests that at least two species of the genus Contracaecum were found (see comments 1 and 2). .

 9. Table 1: giving the intensity for each cormorant separately does not make sense here - this is simply the number of nematodes found (column 3 and 5 are the same).

Please also add to the table i.e. for each cormorant - how many L4 were found and how many females and males were found.

 Column 2: more precisely „N. birds”.

Column 4: should be „location” or „habitat”.

 10. Line 170: the whole name of Contracaecum rudolphii (that is, with author and date) should be at the first mention in the text that is, in the chapter Introduction  (line 40).

 11. Lines 171-172: „Regarding the cormorant, C. rudolphii s.l. was found infecting Ph. carbo in the Czech Republic and Germany [17]…” – very imprecise, incomplete data - the nematode C. rudolphii was also recorded in cormorants in other countries, even if in Poland (the authors cite these works !? - [5] and [7]). This part of the discussion (lines 171-173) needs supplementation.

Author Response

Dear reviewer, many thanks for your comments and we're glad that you liked our revisions.

  1. Title: since nematodes are identified to a species, then you should use this name (Contracaecum rudolphii) in the title. Specifying "Contracaecum sp." is pointless.

R: Dear reviewer, we replaced "Contracaecum sp." with Contracaecum rudolphii according to your precious suggestion.

  1. Simple summary, Abstract: see comment 1. The abbreviation "sp." stands for singular (one unidentified species); the abbreviation "spp." means plural, i.e. at least two species. Please rephrase these two chapters accordingly.

R: Dear reviewer, we rephrase the abbreviation according to your precious suggestions.

  1. Lines 39-40: „Among the Anisakidae family, Contracaecum spp. nematodes occur as adults in the stomach of different piscivorous birds, including pelicans and cormorants [4]. The complex life cycle of Contracaecum rudolphii s.l. includes aquatic invertebrates as first intermediate hosts (small crustaceans, polychaetes and gastropods)” – not clear – These two sentences have no connection with each other - first the genus "Contracaecum" is mentioned and then suddenly the species "C. rudolphii" appears. It would have been appropriate to write that this species of nematode occurs in the cormorant.

R: Dear reviewer, we tried to connect the two sentences according to his suggestion.

  1. Lines 41-42: „… invertebrates as first intermediate hosts (small crustaceans, polychaetes and gastropods) [5],…” – incorrect citation, the authors of this publication Kanarek and Bohdanowicz (2009) did not study the life cycle of C. rudolphii.

R: Dear reviewer, we apologise for the mistake, we change this reference with Moravec (2009 doi: https://doi.org/10.14411/fp.2009.023)

5.Lines 43-45: „Mediterranean, the presence of C. rudolphii s.l. was documented in fish …[6,7]” - incorrect citation, authors of publication [7] studied fish only from Poland.

Dear reviewer, we apologise for the mistake, we change this reference with Farjallah et al. (2008 doi: 10.1016/j.parint.2008.05.003)

  1. Lines 74-76: „A total of four specimens of Phalacrocorax carbo sinensis, the Eurasian subspecies of the Great Cormorant, were collected from 2011 to 2022” – strange notation - four cormorants were collected in 12 years ? Please remove this sentence, and add the year data to the next paragraph (to specific cormorants).

Dear reviewer, we rephrase the sentence according to your precious suggestion.

  1. Lines 101-103: „ …following the taxonomic keys based on the morphology of lips and interlabial tips, length of spicule and morphology of the spicule tip [15]: - incorrect citation, in this article [Li et al., 2005] there is no description of this nematode species.

R: Dear reviewer, we apologise for the mistake, we change this reference with Shamsi et al. (2009 doi: 10.1007/s00436-009-1424-y).

  1. Line 120: since the nematode species was identified, it must be stated that C. rudolphii was found; the name "Contracaecum spp." suggests that at least two species of the genus Contracaecum were found (see comments 1 and 2).

R: Dear reviewer, we rephrase the abbreviation according to your precious suggestions.

  1. Table 1: giving the intensity for each cormorant separately does not make sense here - this is simply the number of nematodes found (column 3 and 5 are the same). Please also add to the table i.e. for each cormorant - how many L4 were found. Column 2: more precisely „N. birds”. Column 4: should be „location” or „habitat”.

R: Dear reviewer, we changed table 1 according to your suggestion.

  1. Line 170: the whole name of Contracaecum rudolphii (that is, with author and date) should be at the first mention in the text that is, in the chapter Introduction (line 40).

R: Dear reviewer, we added the author and date of C. rudolphii in tbe first mention of the introduction as suggested.

  1. Lines 171-172: „Regarding the cormorant, C. rudolphii s.l. was found infecting Ph. carbo in the Czech Republic and Germany [17]…” – very imprecise, incomplete data - the nematode C. rudolphii was also recorded in cormorants in other countries, even if in Poland (the authors cite these works !? - [5] and [7]). This part of the discussion (lines 171-173) needs supplementation.

R: Dear reviewer, we added more information on the presence of C. rudolphii s.l. in the main document according to your suggestion.

Reviewer 2 Report

This manuscript identified Contracaecum rudolphii in the digestive tracts of four natural died great cormorants using morphological and PCR-R, which adds some valuable information in the field. However, there still some problems in the ms needed to clarified. For examples, in the introduction, how many nematode parasites have been detected or named in the great cormorants? If there are only one species: Contracaecum rudolphii, the relationship of Contracaecum rudolphii A, B and i s.l. needs to be added. The results need to supplement photos of larvae and adults. It will enhance the reliability of species of identification.

Author Response

  1. In the introduction, how many nematode parasites have been detected or named in the great cormorants? If there are only one species: Contracaecum rudolphii, the relationship of Contracaecum rudolphii A, B and i s.l. needs to be added

R: Dear reviewer, we added the information about the ecological aspects of Contracaecum rudolphii A and B (lines 44-46).

  1. The results need to supplement photos of larvae and adults. It will enhance the reliability of species of identification.

R: Dear reviewer, we added photos of the morphological identification according to you precious suggestion.

Reviewer 3 Report

Identification of Contracaecum sp. (nematoda: Anisakidae) in 2 great cormorants Phalacrocorax carbo sinensis (Blumenbach, 3 1978) from Southern Italy

The work is valid and I have few suggestions.

The conclusion of the abstract should be in agreement and similar to the general conclusion of the paper.

Keywords should be different from the title.

Titles of tables and figures should be revised. All should be self-explanatory and there should be no abbreviations in these titles.

In figure 3, the columns should also be numbered.

Author Response

  1. The conclusion of the abstract should be in agreement and similar to the general conclusion of the paper.

R: Dear reviewer, we rephrase the conclusion of the abstract according to your precious suggestions.

  1. Keywords should be different from the title.

R: Dear reviewer, we changed the keywords according to your precious suggestion.

  1. Titles of tables and figures should be revised. All should be self-explanatory and there should be no abbreviations in these titles.

R: Dear reviewer, we tried to correct the titles of the figures and tables in order to be much clearer.

  1. In figure 3, the columns should also be numbered.

R: Dear reviewer, we numbered the columns of figure 3 as suggested.

Round 2

Reviewer 1 Report

I have a few more comments.

1. Line 39: To zdanie „The Contracaecum genre includes organisms with a complex life cycle such as Contracaecum rudolphii…” - still sounds strange - after all, all Contracaecum have a complex life cycle.

2. Line 104: The authors write that they replaced the paper by Li et al., 2005 with Shamsi et al. (2009 doi: 10.1007/s00436-009-1424-y) - however, I do not see this paper in the references list. Besides, in the mentioned paper there is a morphological description of C. rudolphii D and C. rudolphii E, while currently the authors stated C. rudolphii A and B ??. Besides, are there really, visible morphoanatomical differences between Contracecum rudolphii A, B, C and D ? - I doubt it; they are distinguishable only at the genetic level. Here, please cite a paper with diagnostic features of C. rudolphii only. After all, here it is about morophanatomical identification of C. rudolphii (not A, B).

3. Figure 2. Photomicrographs of the Contracaecum adults found. a = Mouth region of 141 the aadult form; b = Tail region of the adult form:

Adding photos of nematodes is a good idea. Only these photos do not show diagnostic features: lips – dorsal and ventro-lateral (are of poor quality and you can't see all of them), interlabia (adult stages), intestinal caecum, ventricular appendix, excretory pore, etc., etc. If the authors have good-quality photos of adult stages (males, females) and larvae then it is worth posting them; but these two photos, in my opinion, add nothing to this work.

4. Line 120:  „All specimens of P. carbo sinensis were found to be infected by Contracaecum sp….” – about this I have already written, since the species is identified, here should be C. rudolphii.

5. Table 1. Infestation parameters of the Contracaecum spp. – see comment 4.

6. Lines 180-182: C. rudolphii has been reported in P. carbo also in other countries, e.g. Italy; earlier I wrote about Poland, because the authors cited a paper from Poland. So, you need to supplement this information or (if the authors are not sure about the occurrence of this parasite) rewrite this sentence differently - without countries.

Author Response

  1. Line 39: To zdanie „The Contracaecum genre includes organisms with a complex life cycle such as Contracaecum rudolphii…” - still sounds strange - after all, all Contracaecum have a complex life cycle.

Reply: We thank the reviewer for the comment. The sentence has been modified as follows: “The genus includes organisms with a complex life cycle such, having aquatic invertebrates as first intermediate hosts (small crustaceans, polychaetes and gastropods) [5], whereas fish represent the second intermediate or paratenic hosts. In the Mediterranean, the presence of Contracaecum rudolphii Hartwich, 1964 (s.l.) was…..”

  1. Line 104: The authors write that they replaced the paper by Li et al., 2005 with Shamsi et al. (2009 doi: 10.1007/s00436-009-1424-y) - however, I do not see this paper in the references list. Besides, in the mentioned paper there is a morphological description of C. rudolphii D and C. rudolphii E, while currently the authors stated C. rudolphii A and B ??. Besides, are there really, visible morphoanatomical differences between Contracecum rudolphii A, B, C and D ? - I doubt it; they are distinguishable only at the genetic level. Here, please cite a paper with diagnostic features of C. rudolphii only. After all, here it is about morophanatomical identification of C. rudolphii (not A, B).

Reply: We agree with the reviewer about the misunderstanding. No morphoanatomical differences are visible between Contracecum rudolphii A, B, C and D. Indeed, Two papers can be used to identify at gross morphological level Contracaecum and C. rudolphii, so we have changed the references accordingly:

-Barus, V., Sergeeva, T. P., Sonin, M. D. and Ryzhikov, K. M. (1978). Helminths of Fish-Eating Birds of the Palaeartic Region. I Nematoda. Academia, Prague.

-Fagerholm, H. P. (1991). Systematic implications of male caudal morphology in ascaridoid nematode parasites. Systematic Parasitology 19, 215–228.

  1. Figure 2. Photomicrographs of the Contracaecum adults found. a = Mouth region of 141 the adult form; b = Tail region of the adult form:

Adding photos of nematodes is a good idea. Only these photos do not show diagnostic features: lips – dorsal and ventro-lateral (are of poor quality and you can't see all of them), interlabia (adult stages), intestinal caecum, ventricular appendix, excretory pore, etc., etc. If the authors have good-quality photos of adult stages (males, females) and larvae then it is worth posting them; but these two photos, in my opinion, add nothing to this work.

Reply: we are aware that images are not highly informative for specific characters, however we prefer to keep them in order to disseminate the gross morphology of buccal and caudal regions

  1. Line 120:„All specimens of P. carbo sinensis were found to be infected by Contracaecum sp….” – about this I have already written, since the species is identified, here should be C. rudolphii.

Reply: we have changed with Contracaecum rudolphii (s.l.)

  1. Table 1. Infestation parameters of the Contracaecum spp. – see comment 4.

Reply: we have changed with Contracaecum rudolphii (s.l.)

  1. Lines 180-182: C. rudolphii has been reported in P. carbo also in other countries, e.g. Italy; earlier I wrote about Poland, because the authors cited a paper from Poland. So, you need to supplement this information or (if the authors are not sure about the occurrence of this parasite) rewrite this sentence differently - without countries.

Reply: we have changed sentences as follows, mentioning countries outside Europe before, and then European countries, including Italy. “Regarding the cormorants, to date C. rudolphii (s.l.) has been detected outside Europe, in particular in Ph. brasilianus in Chile [19], Ph. auritus in USA [14]. Regarding European regions, it was reported in Ph. carbo in the Czech Republic, Poland and Germany [17,18]; in Ph. aristotelis in Spain [15,20] and also in Italy [4,7]

Reviewer 2 Report

The quality of the manuscript has been improved based on the suggestions from reviewers. However, there are still some confusions needed to be further clarified.

(1)   Based on the RFLP provided in the manuscript, the two genetic markers (ITS, and rrns) of C. rudolphii A and C. rudolphii B are completely different. Please explain the reason why both were identified as a same species, C. rudolphii.

(2)   In the introduction, C. rudolphii documented in both brackish and freshwater fish was named C. rudolphii s.l., occurred in blackish water fish was named as C. rudolphii A, and found in freshwater fish was regarded as C. rudolphii B. The classification system was very confusing. In the manuscript, they are called as sibling species, which means they are separate species, though they possessed close relationship. How do you think?

(3)   In recent years, the similar results for the Contracaecum rudolphii A and C. rudolphii B from cormorants have been reported from Italy (Mattiucci et al. 2020, 10.1007/s00436-020-06658-8; Carmeno et al 2022, https://doi.org/10.1016/j.vprsr.2021.100674). The present manuscript used the same methods to identity the parasite in 4 cormorants in Southern Italy. Interestingly, the other two published papers concerned the parasite in Central Italy (Mattiucci et al. 2020), and North Italy (Carmeno et al 2022). As a peer researcher, the novelty of the paper of Carmeno et al 2022) and the manuscript are not high. Please permit me to speak plainly.

Other suggestions:

In the title, the species name rudilphii should be italic

Line 38, genre replaced with genera

Table 1. what’s the meaning the “N”? I consider it should not be abbreviated.

In Fig. 2 the characteristics of Contracaecum rudolphii needed to be pointed out.

Lines 181-182, The Ph should be P.

Author Response

The quality of the manuscript has been improved based on the suggestions from reviewers. However, there are still some confusions needed to be further clarified.

(1)   Based on the RFLP provided in the manuscript, the two genetic markers (ITS, and rrns) of C. rudolphii A and C. rudolphii B are completely different. Please explain the reason why both were identified as a same species, C. rudolphii.

Reply: the first identification is based on the gross morphological features and on the host affiliation. Then, the use of molecular markers are useful to distinguish sibling/cryptic species as the case of the member of the C. rudolphii complex. In fact, diagnostic key based on RFLP patterns obtained for ITS and rrnS are detailed in the manuscript D’Amelio, S.; Barros, N.B.; Ingrosso, S.; Fauquier, D.A.; Russo, R.; Paggi, L. Genetic Characterization of Members of the Genus Contracaecum (Nematoda: Anisakidae) from Fish-Eating Birds from West-Central Florida, USA, with Evidence of New Species. Parasitology 2007, 134, 1041–1051, doi:10.1017/S003118200700251X. In this research, specific fixed polymorphisms are described for each of the two subspecies/sibling species of the rudolphii complex, according to sequences analyses and RFLP patterns and supported also by phylogeny. So, according to the general morphology and the genetic characterization, we can confirm that these are specimens belonging to the C. rudolphii complex and in particular to subspecies A and B.

(2)   In the introduction, C. rudolphii documented in both brackish and freshwater fish was named C. rudolphii s.l., occurred in blackish water fish was named as C. rudolphii A, and found in freshwater fish was regarded as C. rudolphii B. The classification system was very confusing. In the manuscript, they are called as sibling species, which means they are separate species, though they possessed close relationship. How do you think?

Reply: we agree with the reviewer that the concept of subspecies/sibling/cryptic species may be easily misleaded. In this regard, we are aware that morphological convergence is a very common phenomenon in endoparasites of this family (Anisakidae) and the identification of genetic distinct groups very closely related at phylogenetic level but with distinct feature (i.e. ecological features or geographical distribution) may occur. For these reasons, we have used sometimes the species name complex (C. rudolphii sl) and when we discussed something more specific we mentioned the subspecies A or B, mostly according to the nomenclature used originally in the reference mentioned.

(3)   In recent years, the similar results for the Contracaecum rudolphii A and C. rudolphii B from cormorants have been reported from Italy (Mattiucci et al. 2020, 10.1007/s00436-020-06658-8; Carmeno et al 2022, https://doi.org/10.1016/j.vprsr.2021.100674). The present manuscript used the same methods to identity the parasite in 4 cormorants in Southern Italy. Interestingly, the other two published papers concerned the parasite in Central Italy (Mattiucci et al. 2020), and North Italy (Carmeno et al 2022). As a peer researcher, the novelty of the paper of Carmeno et al 2022) and the manuscript are not high. Please permit me to speak plainly.

Reply: we agree that previous investigations provide information on C. rudolphii A and B from cormorants. However, geographical region and ecological environment are different. In particular, Carmeno et al reports data from freshwater inner lakes for the per-alpine region (north of Italy), while data from cormorants by Mattiucci et al regards the Orbetello lagoon (central Italy), a brackish water lagoon. The present paper reports data from a variety of environments including freshwater rivers (Platani river), a coastal lake mostly freshwater (Salso lake) as well as marine environments such as Leporano bay and Pantelleria, all from the south of Italy.

Other suggestions:

In the title, the species name rudilphii should be italic

Reply: we have changed it, thanks.

Line 38, genre replaced with genera

Reply: we have changed it.

Table 1. what’s the meaning the “N”? I consider it should not be abbreviated.

Reply: we have added in the table legend: “(N: number of specimens)”

In Fig. 2 the characteristics of Contracaecum rudolphii needed to be pointed out.

Reply: we have modified the sentence in the result section describing better the morphological features corresponding to C. rudolphii sensu Hartwich, 1964, as follows: “The morphological analysis confirmed the presence of 54 fourth stage larvae and 127 adults of Contracaecum rudolphii sensu Hartwich, 1964. Adults showed a brownish-yellowish and transversely striated cuticle, lips divided into two lobes, interlabia well developed and bifurcated in the distal end (Figure 2). Males showed two subequal spicules with a length in a range of 4-10mm, according to the morphological key [15].” Lines 181-182, The Ph should be P.
